# Possible nematic to smectic phase transition in a two-dimensional electron gas at half-filling

Q. Qian[1], J. Nakamura[1], S. Fallahi[1,2], G.C. Gardner[2,3,4] & M.J. Manfra[1,2,3,4,5]

Liquid crystalline phases of matter permeate nature and technology, with examples ranging from cell membranes to liquid-crystal displays. Remarkably, electronic liquid-crystal phases can exist in two-dimensional electron systems (2DES) at half Landau-level filling in the quantum Hall regime. Theory has predicted the existence of a liquid-crystal smectic phase that breaks both rotational and translational symmetries. However, previous experiments in 2DES are most consistent with an anisotropic nematic phase breaking only rotational symmetry. Here we report three transport phenomena at half-filling in ultra-low disorder 2DES: a non-monotonic temperature dependence of the sample resistance, dramatic onset of large time-dependent resistance fluctuations, and a sharp feature in the differential resistance suggestive of depinning. These data suggest that a sequence of symmetry-breaking phase transitions occurs as temperature is lowered: first a transition from an isotropic liquid to a nematic phase and finally to a liquid-crystal smectic phase.

[1] Department of Physics and Astronomy, Purdue University, West Lafayette, IN 47907, USA. [2] Birck Nanotechnology Center, Purdue University, West Lafayette, IN 47907, USA. [3] Station Q Purdue, Purdue University, West Lafayette, IN 47907, USA. [4] School of Materials Engineering, Purdue University, West Lafayette, IN 47907, USA. [5] School of Electrical and Computer Engineering, Purdue University, West Lafayette, IN 47907, USA. Q. Qian and J. Nakamura contributed equally to this work. Correspondence and requests for materials should be addressed to M.J.M. (email: mmanfra@purdue.edu)

A clean two-dimensional electron system (2DES) at half-filling in high-index ($N \geq 2$) Landau levels (LLs) has been theoretically predicted[1] and experimentally observed[2] to enter an anisotropic liquid-crystal phase colloquially deemed the stripe phase. The designation as a stripe phase stems from early theoretical work based on Hartree–Fock analysis that predicted unidirectional charge density wave (CDW) order. Later treatment taking into account quantum fluctuations and disorder raised the possibility of several electronic crystal phases[3]. Candidate states at half-filling are distinguished by their symmetries: the nematic liquid crystal breaks rotational symmetry while preserving translational symmetry[3–6], the smectic liquid crystal (akin to the CDW) breaks rotational symmetry as well as translational symmetry in one spatial direction[3,7–10], and the anisotropic stripe crystal breaks translational symmetry in both directions[3,7,11–15]. At high temperatures or in samples with high disorder, symmetries are restored and the 2DES becomes an isotropic conducting liquid. A qualitative picture of each state[16] is shown in Fig. 1. With multiple candidate states available[17], an intriguing question arises: which symmetries are broken in the limit of low temperature and minimal disorder?

A prominent signature of broken symmetry phases is dramatic transport anisotropy: under typical experimental conditions, the resistance $R_{xx}$ parallel to the $\langle 1\bar{1}0 \rangle$ crystallographic direction of the host gallium arsenide (GaAs) lattice reaches a large peak at half-filling, while the resistance $R_{yy}$ measured parallel to $\langle 110 \rangle$ drops to a low value, indicating that it is much easier for current to flow along the nematic ordering direction rather than perpendicular to it. This orientation can be reversed at high 2DES density[18] or by applying a magnetic field in the plane of the 2DES[19], but it has been found that the easy transport axis is always oriented along one of the crystallographic directions of the GaAs lattice[2,20]. The nature of the native symmetry-breaking mechanism in the lattice remains unknown, although progress has been made towards understanding it[21]. Broken symmetry states are usually observed in high ($N \geq 2$) Landau levels, but exotic methods may be used to induce a transition from a fractional quantum Hall state to an anisotropic phase at half-filling in the $N = 1$ Landau level[22–24]. Previous experiments for $N \geq 2$ showing conductivity saturation at finite values at the lowest temperatures, absence of noise specific to CDW order, and lack of evidence of sharp features in differential resistance measurements have supported the nematic, which exhibits short-range stripe ordering but preserves long-range translational symmetry, as the most likely candidate state at half-filling[4,6,25]. On the other hand, experiments in high-$T_c$ superconducting cuprate materials have shown a series of transitions from isotropic to nematic to smectic as temperature is lowered, with symmetries broken one at a time[26,27].

Here, we investigate an ultra-low disorder GaAs 2DES cooled to millikelvin temperatures, and we report three transport phenomena at half-filling. First, we observe a non-monotonic temperature dependence to the resistance $R_{xx}$, with $R_{xx}$ decreasing as the system is cooled at the lowest temperatures, which is consistent with theoretical predictions for the smectic state. Second, we measure sharp features in the differential resistance suggestive of depinning, which indicates broken translational symmetry. Finally, we observe the onset of time-dependent noise in $R_{xx}$, which is a prominent feature in conventional materials exhibiting CDW order. Our measurements are consistent with the 2DES undergoing a sequence of temperature-induced phase transitions as it is cooled, from an isotropic state to a nematic phase and finally to a smectic phase with broken rotational and translational symmetries.

## Results

**GaAs heterostructures**. A key factor in our experiment is the high quality of our GaAs 2DES. We characterize the sample's quality by its high mobility $\mu = 28 \times 10^6$ cm$^2$ V$^{-1}$ s$^{-1}$ and large energy gap for the fragile $\nu = 5/2$ fractional quantum Hall state, $\Delta_{5/2} = 570$ mK. Figure 2a displays magnetotransport in the $N = 1$ LL; fractional states at $\nu = 5/2$ and $\nu = 12/5$ are well developed. The strength of these fragile states indicates low disorder in the 2DES, and suggests the possibility that fragile phases may emerge in high LLs as well. Henceforth, we refer to this sample as Sample A.

**Temperature dependence of $R_{xx}$**. At low temperature when the magnetic field is tuned near half-filling of the spin-resolved $N = 2$ LL, the 2DES exhibits anisotropic conduction, indicating a phase with broken rotational symmetry. The magnetotransport near $\nu = 9/2$ and $\nu = 11/2$ in Sample A is shown in Fig. 2b, c. The four-terminal resistance $R_{xx}$ measured along the $\langle 1\bar{1}0 \rangle$ direction (the hard transport axis) of the GaAs lattice reaches a large peak near each half-filling, while the resistance $R_{yy}$ measured along the $\langle 110 \rangle$ direction (the easy transport axis) becomes immeasurably small. A central finding of our current work is detailed in Fig. 2d, e: $R_{xx}$ at half-filling has a non-monotonic dependence on temperature. As the system is initially cooled from high temperature, transport becomes anisotropic: $R_{xx}$ increases while $R_{yy}$ drops to nearly zero. This behavior has been observed in previous experiments[2,21,28–30] and has been established as a signature of a phase transition from an isotropic liquid to an anisotropic conducting nematic phase[4]. However, upon further cooling we observe that $R_{xx}$ reaches a peak at intermediate temperature ($T \sim 50$ mK), and then drops without saturating as temperature is lowered further down to $T = 14$ mK; this temperature dependence has not been previously reported, and is not consistent with the expected behavior of a nematic. On the other hand, for the smectic state it is theoretically predicted[12] that the interstripe impurity scattering rate should increase as temperature is lowered

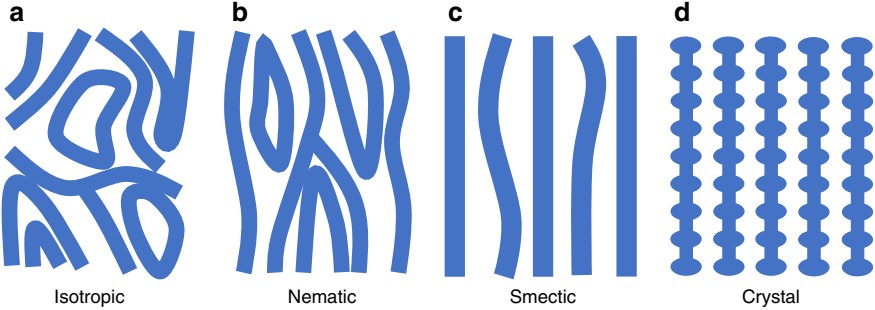

**Fig. 1** Cartoon of electronic liquid-crystal phases at half-filling. **a** Isotropic liquid. **b** Nematic liquid crystal. **c** Smectic liquid crystal. **d** Anisotropic stripe crystal (after Kivelson, Fradkin, and Emery[16]). Blue regions indicate high filling factor (i.e., high density of electron guiding centers) and white regions indicate low filling factor

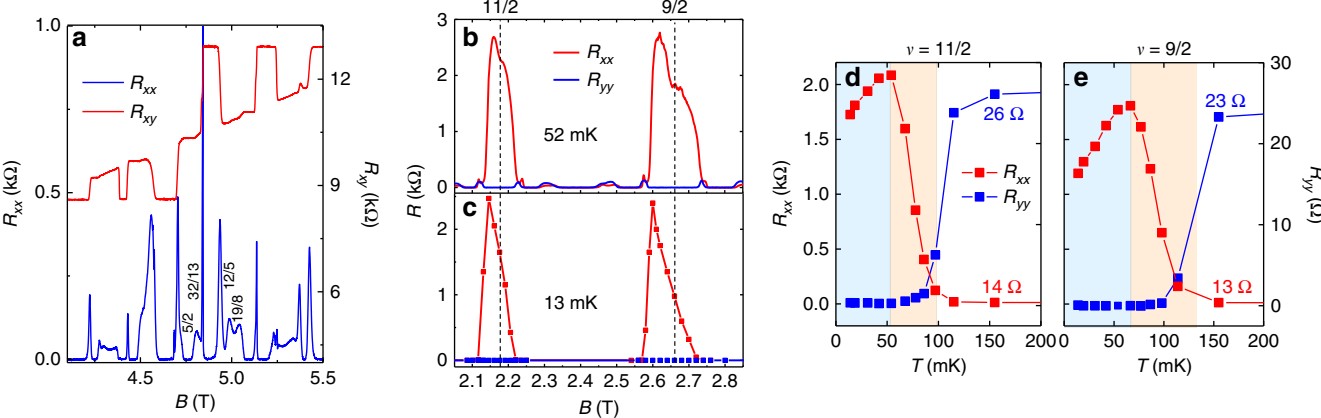

**Fig. 2** Low temperature properties of Sample A. **a** Magnetotransport in the $N = 1$ LL from Sample A at $T = 12$ mK. The presence of FQHS at $\nu = 5/2$ and $12/5$ indicates the high quality of this sample. **b** $N = 2$ LL transport showing the anisotropic phases at $\nu = 11/2$ and $\nu = 9/2$ at $T = 52$ mK obtained by sweeping the magnetic field; at this temperature no significant hysteresis or metastability occurs. **c** $N = 2$ LL transport at $T = 13$ mK. Each data point at this temperature was obtained by fixing the magnetic field and cooling from 200 mK down to 13 mK. $R_{xx}$ (red) is measured along the $\langle 1\bar{1}0 \rangle$ direction of the GaAs lattice and $R_{yy}$ (blue) is measured along the $\langle 110 \rangle$ direction. Dashed lines indicate the positions of exact half-filling. **d** $\nu = 11/2$ and **e** $\nu = 9/2$ $R_{xx}$ (red) and $R_{yy}$ (blue) as a function of temperature taken at exactly half-filling factor for each state. The high temperature values of $R_{xx}$ and $R_{yy}$ are labeled. The colors of the regions indicate the different phases: isotropic liquid (white), nematic liquid crystal (pink), and smectic liquid crystal (blue)

due to Luttinger-liquid effects, leading to a decrease of $R_{xx}$ upon cooling. Thus, the decrease of $R_{xx}$ at low temperatures is consistent with the smectic state, and the turnover in $R_{xx}$ is taken as evidence of a transition from a nematic phase with broken rotational symmetry to a smectic phase which additionally breaks translational symmetry. We observe similar turnover at $\nu = 13/2$ (see Supplementary Note 3 and Supplementary Fig. 2). In Supplementary Note 1 we discuss why the turnover of $R_{xx}$ is not easily explained by the coexistence of domains with different local nematic orientation.

**Differential resistance measurements.** Measurements on Sample A of the differential resistance $\mathrm{d}V_{xx}/\mathrm{d}I$ measured along the hard direction at $\nu = 9/2$, shown in Fig. 3, provide additional evidence for a possible transition from nematic to smectic ordering. At temperatures above the point at which the turnover of $R_{xx}$ occurs, the differential resistance $\mathrm{d}V_{xx}/\mathrm{d}I$ exhibits a zero-bias minimum, which is indicative of a pseudogap in the tunneling density of states predicted at half-filling[8,31] and is consistent with the nematic or smectic phase[2,12,32]. At lower temperature, this zero-bias minimum persists and becomes more pronounced; however, we also observe an additional feature: $R_{xx}$ exhibits a plateau and then a sharp increase before decreasing. This feature is reminiscent of depinning observed when finite bias is applied in the insulating bubble states which occur near quarter filling of $N \geq 2$ LLs[32-34]; similar behavior is also observed in other electronic crystal systems outside of the quantum Hall regime[35]. This feature indicates some degree of pinning along the hard axis at half-filling; the state then slides under further increase in DC drive. The nematic is an unpinned state due to its translational invariance and is thus inconsistent with this signature; the smectic state, however, may be pinned by disorder or at the boundaries[3,4,6,36]. Therefore, the nonlinear IV characteristics observed here at low temperatures can be interpreted as evidence of smectic order at lowest temperatures. To our knowledge such sharp features in the differential resistance have not been previously reported at $\nu = 9/2$. We note that significant noise (far above the background noise in our measurement circuit) is visible in the low temperature traces in Fig. 3; this reflects time-dependent fluctuations in $R_{xx}$.

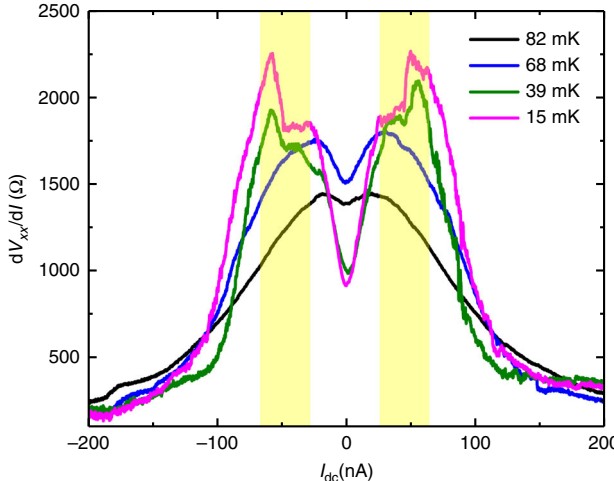

**Fig. 3** Differential resistance vs. DC current in Sample A. The DC current was applied along the hard axis. A 4 nA AC excitation current was also applied to probe $R_{xx}$. The yellow highlighted region indicates where we observe a plateau and then sharp increase in the differential resistance at low temperatures reminiscent of depinning

**Noise measurements.** Next, we characterize these time-dependent resistance fluctuations. Time traces with zero applied DC bias taken at various temperature at $\nu = 9/2$ are shown in Fig. 4a. At $T = 190$ mK, transport is isotropic and we find no excess noise other than the electrical noise in our instruments. As the temperature is lowered and transport becomes anisotropic, noise appears in the measured resistance. The noise amplitude increases as temperature is lowered, and becomes extremely large below the temperature at which the turnover of $R_{xx}$ occurs. In Fig. 4b we show the power spectral density $S_R$ of the resistance noise. At low frequency the noise spectrum falls off approximately as $1/f^\alpha$, where $1 < \alpha < 2$. This suggests that the noise is a combination of a small number of strong two-level fluctuators causing discrete switching (clearly visible in the top trace in Fig. 4a) along with a large number of weak fluctuators with a wide distribution of switching frequencies. See Supplementary Note 2 and Supplementary Fig. 1 for additional discussion of the noise.

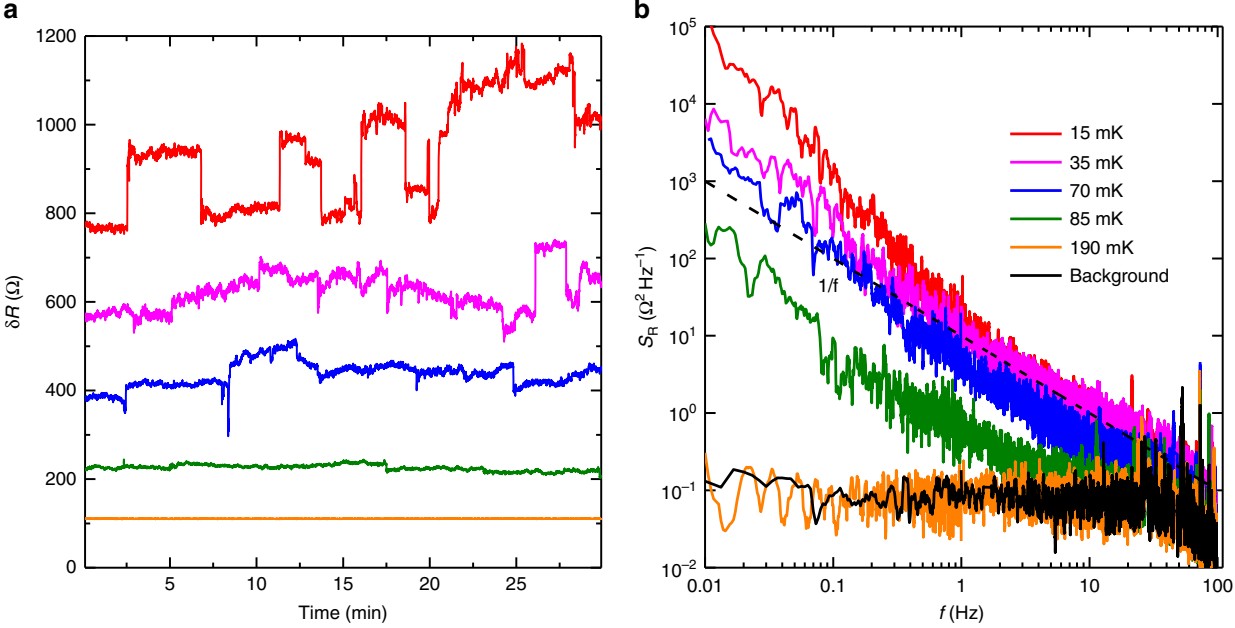

**Fig. 4** Noise measurement of Sample A at various temperatures at $\nu = 9/2$. **a** Time traces showing the fluctuations in $R_{xx}$. **b** Power spectrum $S_R$ of the noise at each temperature. The dashed line indicates a slope of $1/f$. The bottom black trace shows the background noise level in our instruments

At first blush the resistance fluctuations resemble charge noise[37] frequently encountered in mesoscopic devices, but it is remarkable that we observe this type of noise in such a large, macroscopic sample (4 mm × 4 mm) and it increases substantially as temperature is decreased. Noise has been reported in a low LL filling Wigner crystal state[38]; however, we believe our study is the first to report noise at half-filling. Importantly, we do not observe this noise at zero magnetic field or in the vicinity of any quantized Hall states; the noise is clearly related to the electronic phase at half-filling. Low-frequency noise following the trend $1/f^\alpha$, including discrete switching noise, has been widely reported in conventional CDW materials[39], including $NbS_3$,[40] $K_{0.3}MoO_3$,[41] and $TaS_3$;[42] this noise has been explained as being due to the motion of topological defects in the CDW order. We interpret the onset of noise at half-filling at low temperature as additional evidence for CDW ordering and a transition to the smectic phase. Additionally, we note that onset of noise does not exactly coincide with the turnover of $R_{xx}$; this suggests that there is a smooth crossover rather than an abrupt phase transition from nematic to smectic order. The smoothness of this transition could be due to finite disorder (present even in our cleanest sample), or might be inherent to this type of transition in quasi two-dimensional systems.

An alternative mechanism for noise has been proposed in which resistance fluctuations are caused by thermally populated local variations in nematic order[43]. On its face, our data seem inconsistent with this mechanism since thermally excited fluctuations should decrease as temperature is lowered. On the other hand, an unusual order-by-disorder situation favoring a more disordered state at lower temperature might enable this mechanism to yield noise that increases as the 2DES is cooled; our data do not rule this possibility out entirely.

**Impact of disorder**. To investigate the effect of disorder on the possible nematic to smectic phase transition, we have repeated the field cooling and noise measurements on a higher disorder sample from a wafer with the same GaAs/aluminium gallium arsenide (AlGaAs) heterostructure design; we refer to this more disordered wafer as Sample B. We quantify the higher disorder of Sample B by its substantially lower mobility $\mu = 6 \times 10^6$ cm$^2$ V$^{-1}$ s$^{-1}$ and lower $\nu = 5/2$ energy gap $\Delta_{5/2} = 120$ mK. The easy axis in Sample B is also oriented along the $\langle 110 \rangle$ crystallographic direction. We find that at $\nu = 9/2$, Sample B exhibits a smaller reduction in $R_{xx}$ from its peak value with decreasing temperature compared to Sample A, and the temperature at which the resistance turnover occurs is lower than in Sample A, as shown in Fig. 5a. The behavior observed in Sample A is dramatically weakened in the more disordered sample. At $\nu = 11/2$, Sample B shows a monotonic increase of $R_{xx}$ as temperature is lowered and exhibits no turnover at all; this may indicate spin dependence to the strength of the smectic state, since it appears that disorder has destroyed the smectic at 11/2, but only weakened it at 9/2.

Figure 5c shows the noise amplitude defined as the integral of the spectral density $\Delta R \equiv \sqrt{2 \int_{0.01\text{Hz}}^{100\text{Hz}} S_R(f) df}$ vs. $1/T$ for Sample A and Sample B. Sample A shows substantial noise at $\nu = 9/2$ and $\nu = 11/2$. While Sample B also shows noise at low temperature at $\nu = 9/2$, it is substantially smaller than for Sample A, and Sample B shows no excess noise at $\nu = 11/2$. We conclude that two of the signatures of possible smectic order (turnover of $R_{xx}$ and conductance noise) are substantially weaker in the higher disorder (but otherwise nearly identical) Sample B at $\nu = 9/2$ and absent at $\nu = 11/2$. We measured another sample with intermediate mobility which is consistent with this trend (see Supplementary Note 4 and Supplementary Fig. 3). This indicates that the phenomenon observed here is quite fragile and easily weakened or destroyed by disorder, as might be expected for the highly ordered smectic state. This may explain why the data discussed here have not been observed previously: extremely high-quality 2DEGs and electron temperatures well below $T = 30$ mK are simultaneously required.

We have presented experimentel evidence for a sequence of temperature-induced phase transitions at half-filling, first from an isotropic liquid to a nematic state breaking rotational symmetry and then from the nematic to a smectic state breaking translational symmetry along one axis. An intriguing question that remains is whether the remaining translational symmetry

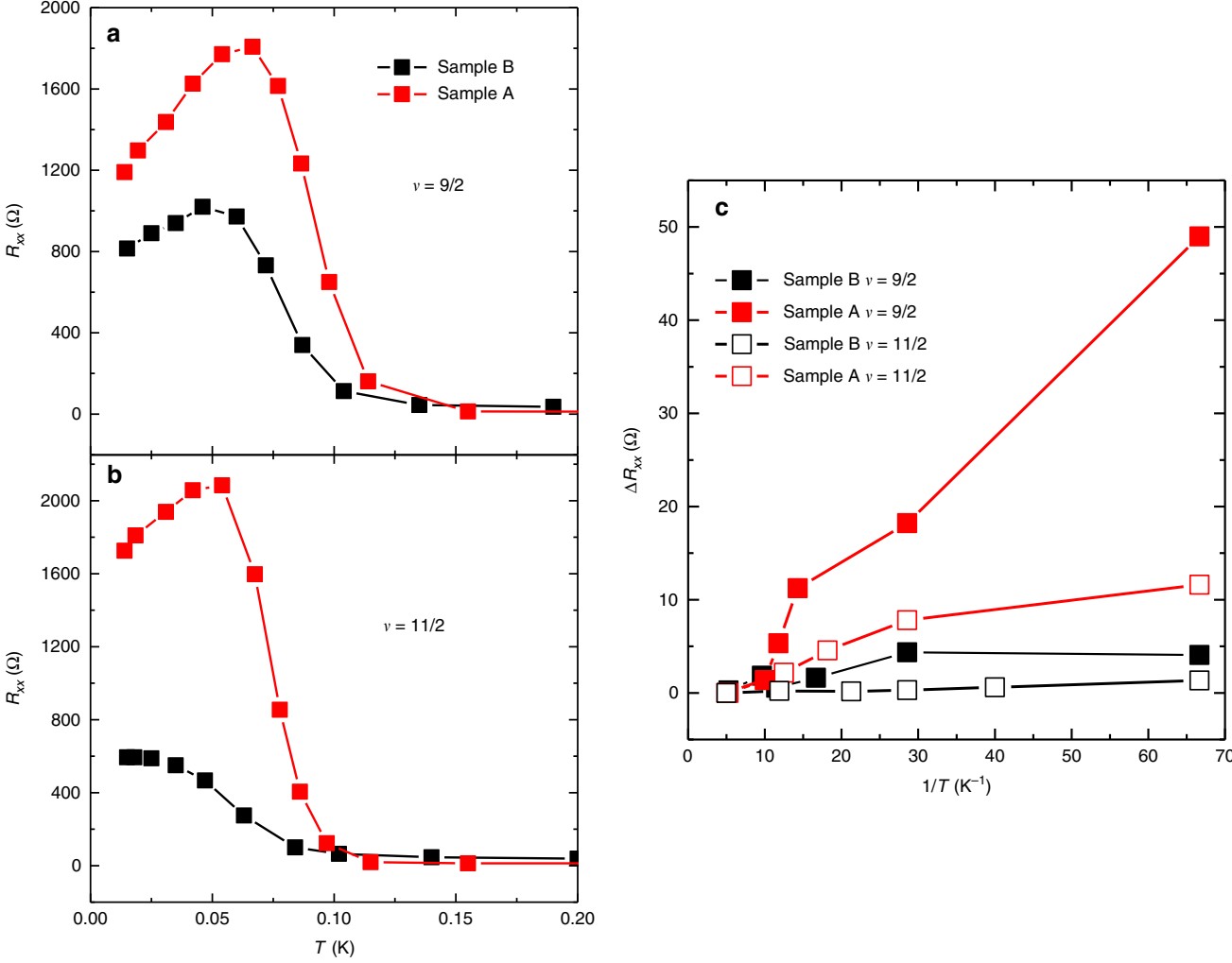

**Fig. 5** Impact of disorder on turnover and noise in $R_{xx}$. Field cooling of Sample A (red) and Sample B (black) at **a** $\nu = 9/2$ and **b** $\nu = 11/2$. Sample A has extremely low disorder, while Sample B has significantly higher disorder. **c** Noise amplitude for samples A and B at $\nu = 9/2$ (open squares) and $\nu = 11/2$ (filled squares)

parallel to the stripes can be broken by a phase transition to an insulating stripe crystal state, which is another candidate state at half-filling[3,7,11,12,14]. We note that $R_{xx}$ shows no sign of saturating at a finite value in our data, which leaves open the possibility that a fully insulating anisotropic stripe crystal breaking all spatial symmetries may develop if even lower electron temperatures can be achieved.

## Methods

**Heterostructure design**. The wafers used in these studies are symmetrically doped GaAs/AlGaAs heterostructures grown by molecular beam epitaxy. The 2DES resides in a 30 nm GaAs quantum well flanked by symmetric AlGaAs barriers and Si doping wells; this doping scheme has been found to produce the highest quality transport in the quantum Hall regime. The samples are 4 mm × 4 mm Van der Pauw squares with electron density $n \approx 2.9 \times 10^{11}$ cm$^{-2}$, and are illuminated with a red LED at $T \approx 10$ K prior to measurement.

**Measurement techniques**. Experiments were performed in a dilution refrigerator with base temperature $T \approx 12$ mK. Our measurement leads have been filtered and well thermalized to the mixing chamber to ensure that the electron temperature in our sample reaches the mixing chamber temperature. The magnetic field is applied perpendicular to the 2DES plane. Measurements are made using standard four-terminal lock-in amplifier techniques using a 4 or 8 nA 435 Hz excitation. NF-LI75 voltage pre-amplifiers are used in conjunction with the lock-in amplifiers to achieve a lower noise level; the pre-amplifiers have an input noise level of $\approx 2.2$ nV Hz$^{-\frac{1}{2}}$.

The same AC lock-in amplifier techniques were used for the resistance noise measurements. An 8 nA 435 Hz excitation was used. The time constant of the lock-in amplifier was set to its minimum value of 1 ms for these measurements, which ensures that the noise is not filtered in the frequency range we present. The output of the lock-in amplifier was connected to a National Instruments NI-9239 24-bit digitizer with 2 kS s$^{-1}$ sampling rate. The background noise in these measurements is limited by the voltage pre-amplifiers to $\approx 2.2$ nV Hz$^{-\frac{1}{2}}$.

**Data availability**. Data available on request from the authors.

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

## Acknowledgements

This work was supported by the Department of Energy, Office of Basic Energy Sciences, under Award number DE-SC0006671. Additional support for sample growth from the W.M. Keck Foundation and Microsoft Station Q is gratefully acknowledged. M.J.M. acknowledges insightful discussions with G.A. Csathy, S.H. Simon, J.P. Eisenstein, I. Sodemann, B. Rosenow, and E. Carlson. We thank J.P. Eisenstein for helpful comments on this manuscript.

## Author contributions

Q.Q., J.N. and M.J.M. conceived the experiment. S.F., G.C.G., and M.J.M. grew the GaAs/AlGaAs wafers. Q.Q., J.N. and M.J.M. performed the measurements and analyzed the data. The manuscript was written by J.N., Q.Q. and M.J.M. with input from all authors.
