## [Peer Review File · Nature Communications]

Reviewers' comments:

Reviewer #1 (Remarks to the Author):

The manuscript reports low temperature measurements on the magnetotransport properties of the collective electronic liquid crystalline phases appearing at half filling of high Landau levels in ultra-clean two-dimensional electron systems. In particular the authors report evidence for the existence of an altogether new liquid crystalline quantum state, i.e. an electronic smectic, appearing in at low temperatures.

I find these new results and interpretation fascinating and the data and arguments supporting them very convincing. The existence of novel liquid crystalline phases, beyond nematic, in high order half-filled Landau levels has been predicted theoretically, but until this work no conclusive experimental evidence supported their existence in real systems. As the authors point out, this is likely due to the high quality quantum well samples that they are able to grow. Furthermore, the fact that the authors see consistent evidence for the existence of this smectic phase not only in the temperature dependence of the longitudinal resistance (as expected by the theory, see ref. 12), but also in the differential resistance and time-dependent fluctuations of the resistance in the sample. In my opinion, these data demonstrate the appearance of a new phase and their interpretation of this phase as a smectic is backed up by theory and is plausible.

Moreover, the authors have been careful to rule out common, and mundane, effects that could possibly complicate their interpretation, in particular by field cooling their sample and checking for hysteresis in the longitudinal resistance.

It is my recommendation that this manuscript be accepted for publication in Nature Communications after a couple of minor points are addressed. I list these below:

1. It seems that the noise in the resistance data onsets at a temperature below in the region where the system is thought to be in a nematic state. In fact it seems that the noise, albeit smaller, is present even in the region where the resistance in the hard-direction is still increasing. How do the author understand this behavior? It would be helpful to the reader to include a brief statement explaining this feature of the data.
2. When discussing the impact of disorder, the authors show that at $11/2$ in sample B there is no evidence for a reduction in R_{xx} as the temperature is lowered, in contrast to $9/2$ in the same sample. They go on to state that this may indicate spin dependence to the strength of the smectic state. Could this difference not also be attributable to disorder? If so, the authors should add a statement to this effect.

Reviewer #2 (Remarks to the Author):

The authors present state of the art measurements of the electronic phases of the fractional quantum Hall regime. They find anomalous temperature dependence in resistivity consistent with isotropic liquid-nematic-smectic transitions.

Overall these results are new and very interesting. It seems that a series of transitions are occurring that need further understanding and exploration.

The manuscript is well written and opens the door to exploring interesting correlated states of electrons. I recommend publication in Nature Communications provided the authors address the following comments:

A central part of their work is the 200-600 Ohm upturns in R_{xx} vs. T shown in Fig. 1 and elsewhere. However, Fig. 3 shows that R_{xx} can fluctuate by 200-400 Ohms on long time scales. I

would like the authors to clarify the time sequence for which they took the data in Fig. 1. Is it possible that the results in Fig. 1 are a short time (and therefore misleading) average in one of the temporal noise plateaus?

I agree that a low T smectic state is consistent with the observations. However, I think the authors should leave room for other possibilities. In the "Noise" section they mention the possibility of a series of charge domains at low T but rule it out because of the temperature dependence where they state: "We note that our observations are inconsistent with a model of thermally-populated local variations in nematic order [41], since thermally excited fluctuations should decrease as temperature is lowered." Is it possible that an order by "disorder" mechanism occurs where the entropy in the free energy starts to favor an ordered state as T is increased? Can the authors rule out a thermal order-by-disorder mechanism?

Can the authors cite other FQHE experiments finding the time-dependent noise of the type shown in Fig. 3 or is this the first of its kind?

It appears that the notation for differential resistance conflicts with the notation for conventional longitudinal resistance. Is it more technically correct to leave the differential resistance as dV_{xx}/dI in Fig. 2 or is it the same as R_{xx} in Fig. 1?

Reviewer #3 (Remarks to the Author):

In this manuscript Qian et al., show experimentally different phase transitions occurring at half filling for an electronic liquid crystal. They relate transport signatures (e.g., decrease of R_{xx} as a function of temperature, development of additional features on the differential resistance and noise measurements) at half filling to the existence of a smectic phase in the electronic liquid. The data and scientific methods are sound and it is clear to me that this is an important subject to be explored and that this is a very complete study. However, even when the data is clear the explanation of smectic, nematic and insulating stripe crystal states needs to be revisited, as the central point of the paper some effort should be given to explain this in the clearest way. I recommend this manuscript for its publication after some revisions of the following points are made.

1. «The nematic liquid crystal breaks rotational symmetry while preserving translational symmetry, the smectic liquid crystal (akin to the CDW) breaks rotational symmetry as well as translational symmetry in one spatial direction, and the anisotropic stripe crystal breaks translational symmetry in both directions». As I said before this needs to be re-written to give to the reader a good idea of how these phases are. I strongly believe that a schematic or a cartoon of these phases could help a lot to understand the impact and importance of this study.
2. What is the understanding for the observation of these anisotropic phases only at $N \geq 2$?
3. In Figure 1 d and e it is not clear that transport around $9/2$ and $11/2$ is isotropic at high temperature, maybe labeling the value at high temperature could help.
4. A discussion of why other studies in high mobility samples (for example: Du et al., Solid State Communications 109 (1999) 389–394) do not observe the smectic phase is needed.
5. Maybe it is related to the fact that I still can't make a clear idea in my head of the smectic phase but I do not understand why the inter-stripe impurity scattering rate should increase as temperature is lowered due to Luttinger-liquid effects, leading to a decrease of R_{xx} upon cooling.
6. Why the breaking of the translation symmetry gives a decrease in R_{xx} and it doesn't seem to have an effect on R_{yy} ?
7. Why the smectic phase is expected to be pinned by disorder or at the boundaries?
8. Color coding the different phases in figure 1 d and e will help to its understanding.
9. I don't understand why it seems like R_{yy} at 13 mK (Figure 1-c) is zero for all the magnetic field values. If this is not the case and it is only a scaling problem I propose to multiply R_{yy} by a factor

(let's say 10) and label it so it is clear that R_{yy} exists for other values of B (as it is the case for Fig; 1-b)

10. It will be useful to include in figure 2 the noise measurements at other filling factors (where the noise is not observed).

11. Sample B, with a considerably lower mobility, does show a decrease of the R_{xx} value at lower temperature but it doesn't show a large noise amplitude. Does it mean that this noise may not be a fingerprint of the smectic phase?

REFEREE A

1. It seems that the noise in the resistance data onsets at a temperature below in the region where the system is thought to be in a nematic state. In fact it seems that the noise, albeit smaller, is present even in the region where the resistance in the hard-direction is still increasing. How do the author understand this behavior? It would be helpful to the reader to include a brief statement explaining this feature of the data.

This observation may indicate smooth crossover behavior rather than a sharp phase transition from nematic to smectic order. The smoothness of this transition could be due to finite disorder (present even in our cleanest sample), or might be inherent to this type of transition in a quasi two-dimensional system. We agree that this an important point, and have added the following statements at the end of the 2nd paragraph in the “Noise Measurements” section of the manuscript:

Additionally, we note that the onset of noise does not exactly coincide with the turnover of R_{xx} . The smoothness of this transition could be due to finite disorder (present even in our cleanest sample), or might be inherent to this type of transition in quasi two-dimensional systems.

2. When discussing the impact of disorder, the authors show that at $11/2$ in sample B there is no evidence for a reduction in R_{xx} as the temperature is lowered, in contrast to $9/2$ in the same sample. They go on to state that this may indicate spin dependence to the strength of the smectic state. Could this difference not also be attributable to disorder? If so, the authors should add a statement to this effect.

We agree that disorder may be responsible for the absence of turnover at $\nu = 11/2$ in sample B. Our data indicates that the higher level of disorder in sample B fully suppresses turnover and noise at $\nu = 11/2$, but only partially suppresses them at $\nu = 9/2$. We believe this suggests that the energy scale associated with the nematic to smectic transition is

smaller in the upper spin branch compared to the lower spin branch of the $N = 2$ Landau level; however, we do not have an explanation for why the different spins might be different. We have elaborated on this in the revised text. The relevant passage, at the end of the 1st paragraph in the “Impact of disorder” section of the manuscript, now reads as follows:

The behavior observed in Sample A is dramatically weakened in the more disordered sample. At $\nu = 11/2$, Sample B shows a monotonic increase of R_{xx} as temperature is lowered and exhibits no turnover at all; this may indicate spin dependence to the strength of the smectic state, since it appears that disorder has destroyed the smectic at $11/2$, but only weakened it at $9/2$.

REFEREE B

1. A central part of their work is the 200-600 Ohm upturns in R_{xx} vs. T shown in Fig. 1 and elsewhere. However, Fig. 3 shows that R_{xx} can fluctuate by 200-400 Ohms on long time scales. I would like the authors to clarify the time sequence for which they took the data in Fig. 1. Is it possible that the results in Fig. 1 are a short time (and therefore misleading) average in one of the temporal noise plateaus?

The points shown in Fig. 1 are taken as the average of 30 minutes time traces (the same time scale as Fig. 3); we believe that this is long enough to ensure that the time-dependent fluctuations do not significantly confound the measurement (we mention that the 30 minutes time traces are taken after allowing the electrons to thoroughly thermalize at each temperature). Additionally, at the lowest temperature we have taken data over longer time scales (~ 10 hours) and found that although the time-dependent fluctuations persist, there is no significant drift in the average resistance over time.

2. I agree that a low T smectic state is consistent with the observations. However, I think the authors should leave room for other possibilities. In the “Noise” section they mention the possibility of a series of charge domains at low T but rule it out because of the temperature dependence where they state: “We note that our observations are inconsistent with a model of thermally-populated local variations in nematic order [41], since thermally excited fluctuations should decrease as temperature is lowered.” Is it possible that an order by “disorder” mechanism occurs where the entropy in the free energy starts to favor an ordered state as T is increased? Can the authors rule out a thermal order-by-disorder

mechanism?

While we note that systems that exhibit order-by-disorder tend to be rare in nature and we are not aware of any theoretical predictions that such a situation might arise in half filled Landau levels, we agree that we cannot rule this possibility out entirely. Therefore, we have revised the manuscript to acknowledge this possibility. The revised manuscript contains the following paragraph added at the end of the Noise Measurements section:

An alternative mechanism for noise was proposed in Ref. 43, in which resistance fluctuations are caused by thermally-populated local variations in nematic order. On its face, our data seems inconsistent with this mechanism since thermally excited fluctuations should decrease as temperature is lowered. On the other hand, an unusual “order by disorder” situation favoring a more disordered state at lower temperature might enable the mechanism in Ref. 43 to yield noise that increases as the 2DES is cooled; our data does not rule this possibility out entirely.

3. Can the authors cite other FQHE experiments finding the time-dependent noise of the type shown in Fig. 3 or is this the first of its kind?

We mention that the integer and most fractional quantum Hall states are gapped and the bulk becomes incompressible at low temperatures, making them unlikely to exhibit noise. As such, there are few experiments measuring noise in the quantum Hall regime in bulk samples. We have cited Cooper *et al.*, PRL 90, 226803 (2003), Ref. 33 in the manuscript, in which narrow-band noise was observed in the bubble states. We agree that more context is useful for readers, so we have added a citation to Li *et al.*, PRL 67, 1630-1633 (1991), Ref. 38 in the revised manuscript, in which broad-band noise was observed in a low-filling Wigner crystal phase. However, we believe our study is the first to report noise at half filling.

4. It appears that the notation for differential resistance conflicts with the notation for conventional longitudinal resistance. Is it more technically correct to leave the differential resistance as dV_{xx}/dI in Fig. 2 or is it the same as R_{xx} in Fig. 1?

dV_{xx}/dI is indeed more technically correct. We have changed to this notation in the current version of the manuscript and in Fig. 4.

REFeree C

1. \ll The nematic liquid crystal breaks rotational symmetry while preserving translational symmetry, the smectic liquid crystal (akin to the CDW) breaks rotational symmetry as well as translational symmetry in one spatial direction, and the anisotropic stripe crystal breaks translational symmetry in both directions \gg . As I said before this need to be re-written to give to the reader a good idea of how this phases are. I strongly believe that a schematic or a cartoon of these phases could help a lot to understand the impact and important of this study.

In order to give the readers a clear picture of the different possible electronic phases, in the new version of our manuscript we have included a cartoon of these phases as the new Fig. 1.

2. What is the understanding for the observation of these anisotropic phases only at $N \geq 2$?

The key difference between the $N = 0$ Landau level and excited Landau levels is that excited Landau levels possess nodes in their wavefunction. These nodes reduce the Coulomb repulsion at short ranges, making it favorable to form clustered states (the nematic/smectic state at half filling and bubble phases on the flanks). The $N = 1$ Landau level, having a single node, is an intermediate case in which a stripe phase may form under exotic conditions (see Samkharadze *et al.*, Nature Physics 12, 191-196 (2016), Ref. 24 cited in the manuscript). See Koulakov, Fogler, and Shklovskii, PRL 76, 499-502 (1996) (Ref. 1 in the manuscript) for a more rigorous analysis based on Hartree-Fock theory.

3. In Figure 1 d and e it is not clear that transport around $9/2$ and $11/2$ is isotropic at high temperature, maybe labeling the value at high temperature could help.

Following this suggestion, we have updated this figure to include a label of the high temperature ($T \sim 150$ mK) values of R_{xx} and R_{yy} .

4. A discussion of why other studies in high mobility samples (for example: Du *et al.*, Solid State Communications 109 (1999) 389-394) do not observe the smectic phase is needed.

We believe there are two factors that have limited observation of the smectic state in previous experiments: sample quality and *electron* temperature. Quality of the best available GaAs samples has improved dramatically since the publication of the paper by Du *et al.* in 1999. An additional consideration is that achieving low electron temperatures is much more

difficult than reaching low mixing temperature temperatures; thus, previous experiments reporting fridge temperatures below 30 mK may have had much higher electron temperatures, and thus been too warm to stabilize the smectic state. Our dilution refrigerator has been modified to achieve large surface area for thermal contact of the measurement leads to mixing chamber, enabling us to achieve very low electron temperatures. We have mentioned in the text that we believe sample quality and electron temperature are the key parameters. This discussion, located at the end of the 2nd paragraph in the section “Impact of disorder”, reads as follows:

We conclude that two of the signatures of smectic order (turnover of R_{xx} and conductance noise) are substantially weaker in the higher disorder (but otherwise nearly identical) Sample B at $\nu = 9/2$ and absent at $\nu = 11/2$. We measured another sample with intermediate mobility which is consistent with this trend (see Supplementary Information section IV and Supplementary Fig. 3). This indicates that the phenomenon observed here is quite fragile and easily weakened or destroyed by disorder, as might be expected for the highly ordered smectic state. This may explain why the data discussed here has not been observed previously: extremely high quality 2DEGs and electron temperatures well below $T = 30$ mK are simultaneously required.

5. Maybe it is related to the fact that I still cant make a clear idea in my head of the smectic phase but I do not understand why the inter-stripe impurity scattering rate should increase as temperature is lowered due to Luttinger-liquid effects, leading to a decrease of R_{xx} upon cooling.

The smectic state consists of one-dimensional electron stripes, and the one-dimensional nature allows this state to be treated with a Luttinger liquid model. In the theoretical work by MacDonald and Fischer (Phys. Rev. B 61, 5724; Ref. 12 in the manuscript) the smectic is treated as a chiral Luttinger liquid, with left and right moving chiral edge channels occuring on opposite edges of each stripe.

A property of interacting Luttinger liquids is that the tunneling density of states is suppressed at low energy (this is qualitatively different from a 2-D Fermi liquid). This suppressed density of states leads to a *decrease* of forward tunneling or, equivalently, *increased* backscattering by an impurity potential barrier at low temperature. A more rigorous derivation of this result can be obtained from a renormalization group transformation. For a useful review of Luttinger liquid theory, see M. P. Fischer and L. I. Glazman, arXiv:cond-

mat/9610037. In the smectic state the left-moving channel and right-moving channels occur on opposite edges of each stripe, so backscattering sends charge across a stripe.

6. Why the breaking of the translation symmetry gives a decrease in of R_{xx} and it doesnt seem to have an effect on R_{yy} ?

In the theoretical work by MacDonald and Fisher (Ref. 12 in the text), it is predicted that ρ_{yy} should in fact increase as temperature is lowered. In our experiment, however, R_{yy} at $9/2$ and $11/2$ becomes immeasurably small; it is possible that R_{yy} does increase, but remains too small for us to measure. At $\nu = 13/2$, however, we *do* observe an increase in R_{yy} as temperature is lowered and we have a discussion of this effect in the supplementary material section III.

7. Why the smectic phase is expected to be pinned by disorder or at the boundaries? If the system had perfect translational symmetry, then there would be no energy cost for translational displacement of the smectic, and it could slide freely. However, in a real sample translational symmetry is broken by the boundaries and also by the random impurity potential; thus there will be some position of of the smectic order that minimizes the system energy, and it will be pinned against translational motion away from this lowest energy position. Conventional charge density wave materials exhibit pinning (and corresponding sharp depinning features in the nonlinear resistance when voltage bias is applied). G. Gruner, Rev. Mod. Phys. 60, 1129-1181 (1988), Ref. 39 in the manuscript, contains a thorough review of pinning in CDW systems.

8. Color coding the different phases in figure 1 d and e will help to its understanding.

We thank referee C for this suggestion. In the new version of our manuscript, the color coding is added to distinguish different phases.

9. I dont understand why it seems like R_{yy} at 13 mK (Figure 1-c) is zero for all the magnetic field values. If this is not the case and it is only a scaling problem I propose to multiply R_{yy} by a factor (lets say 10) and label it so it is clear that R_{yy} exists for other values of B (as it is the case for Figure 1-b).

At $T = 13$ mK, the anisotropic smectic state is immediately flanked by the incompressible bubble states. R_{yy} is immeasurably small in both states; in fact, it is smaller than the noise level of our instruments ($\approx 0.3 \Omega$).

10. It will be useful to include in figure 2 the noise measurements at other filling factors (where the noise is not observed).

We have included a version of the noise figure with this data included, see Fig. 1 of this response. However, we feel that this figure already contains a large amount of information, and adding additional data not directly related to the stripes may confuse readers, so we have not included this in the manuscript.

FIG. 1. From red line to orange line: time traces showing the fluctuations in R_{xx} at $\nu = 9/2$ at different temperatures. Black line: time trace of R_{xx} measured at a compressible state away from half filling at $T = 15$ mK.

11. Sample B, with a considerably lower mobility, does show a decrease of the R_{xx} value at lower temperature but it doesn't show a large noise amplitude. Does it mean that this noise may not be a fingerprint of the smectic phase?

In Sample B, at $\nu = 9/2$ both noise and turnover of R_{xx} are present, but smaller compared to Sample A. Furthermore, in Sample B both effects are entirely absent at $\nu = 11/2$. From our data it appears that the noise and turnover of R_{xx} occur together. We believe this

suggests that the noise is also a signature of the smectic phase.

REVIEWERS' COMMENTS:

Reviewer #1 (Remarks to the Author):

The authors have satisfactorily addressed all the reviewer comments, questions and suggestions. It is my opinion that this manuscript has been sufficiently revised and I recommend publication in Nature Communications.

Reviewer #2 (Remarks to the Author):

All three referees have praise for the paper and all have made comments to help improve the manuscript. The authors have appropriately included all referee comments.

I strongly recommend that the manuscript is published.

Reviewer #3 (Remarks to the Author):

I have reviewed the manuscript by Prof. Manfra and co-workers. The last version of the manuscript showed great improvement and the answers to the reviewers' comments and questions have been addressed in a positive way. I therefore recommend this manuscript for its publication in Nature Communications.